# Dendritic Cell-Derived Artificial Microvesicles Inhibit RLS_40_ Lymphosarcoma Growth in Mice via Stimulation of Th1/Th17 Immune Response

**DOI:** 10.3390/pharmaceutics14112542

**Published:** 2022-11-21

**Authors:** Oleg V. Markov, Aleksandra V. Sen’kova, Islam S. Mohamed, Elena V. Shmendel, Mikhail A. Maslov, Anastasiya L. Oshchepkova, Evgeniy V. Brenner, Nadezhda L. Mironova, Marina A. Zenkova

**Affiliations:** 1Institute of Chemical Biology and Fundamental Medicine SB RAS, Lavrentieva Ave. 8, 630090 Novosibirsk, Russia; 2M.V. Lomonosov Institute of Fine Chemical Technologies, MIREA—Russian Technological University, Vernadskogo Ave. 86, 119571 Moscow, Russia

**Keywords:** dendritic cells, cytochalasin B-induced membrane vesicles, cationic liposomes, murine lymphosarcoma, immune checkpoints, antitumor vaccines, antitumor immunotherapy

## Abstract

Cell-free antitumor vaccines represent a promising approach to immunotherapy of cancer. Here, we compare the antitumor potential of cell-free vaccines based on microvesicles derived from dendritic cells (DCs) with DC- and cationic-liposome-based vaccines using a murine model of drug-resistant lymphosarcoma RLS_40_ in vivo. The vaccines were the following: microvesicle vaccines—cytochalasin B-induced membrane vesicles (CIMVs) obtained from DCs loaded with total tumor RNA using cholesterol/spermine-containing cationic liposomes L or mannosylated liposomes ML; DC vaccines—murine DCs loaded with total tumor-derived RNA using the same liposomes; and liposomal vaccines—lipoplexes of total tumor-derived RNA with liposomes L or ML. Being non-hepatotoxic, CIMV- and DC-based vaccines administered subcutaneously exhibited comparable potential to stimulate highly efficient antitumor CTLs in vivo, whereas liposomal vaccines were 25% weaker CTL inducers. Nevertheless, the antitumor efficiencies of the different types of the vaccines were similar: sizes of tumor nodes and the number of liver metastases were significantly decreased, regardless of the vaccine type. Notably, the booster vaccination did not improve the overall antitumor efficacy of the vaccines under the study. CIMV- and DC- based vaccines more efficiently than liposome-based ones decreased mitotic activity of tumor cells and induced their apoptosis, stimulated accumulation of neutrophil inflammatory infiltration in tumor tissue, and had a more pronounced immunomodulatory activity toward the spleen and thymus. Administration of CIMV-, DC-, and liposome-based vaccines resulted in activation of Th1/Th17 cells as well as the induction of positive immune checkpoint 4-1BBL and downregulation of suppressive immune checkpoints in a raw PD-1 >>> TIGIT > CTLA4 > TIM3. We demonstrated that cell-free CIMV-based vaccines exhibited superior antitumor and antimetastatic activity in a tumor model in vivo. The obtained results can be considered as the basis for developing novel strategies for oncoimmunotherapy.

## 1. Introduction

DCs are professional antigen-presenting cells that trigger antitumor immune responses by the presentation of tumor antigens to T cells accompanied by their stimulation by membrane (co-stimulatory) and soluble (cytokines) molecules. The application of DCs loaded ex vivo with tumor antigens for the activation of antitumor immunity has been intensively investigated in experimental animals and clinical trials for more than 30 years. It has been demonstrated that antitumor DC-based vaccines are highly efficient in various murine tumor models and human tumor xenografts in immunodeficient mice, being administered as monotherapy [1,2] or along with other immunotherapeutics [3,4,5]. Clinical trials have revealed the promising efficiency of DC-based vaccines that are characterized by activation of antitumor T cell immune response and increased overall survival of patients [6,7,8]. However, the preparation of cellular DC-based vaccines is a complex process requiring costly, good manufacturing practices (GMP) and complicated standardization [9] and also have problems with prolonged storage [10].

The application of cell-free vaccines can solve many of the listed problems. Among different antitumor approaches using cell-free technologies, application of extracellular vesicles including exosomes seems to be the most promising. Indeed, DC-derived extracellular vesicles (DEXs) carry key immunostimulatory molecules needed for the activation of antitumor T cell immune response, including MHC class I and II molecules and CD40, CD80, and CD86 co-stimulatory molecules [11]. Hence, DEXs possess the intrinsic ability of DCs to present foreign antigens to T lymphocytes [12]. Furthermore, the lipid bilayer composition of DEXs facilitates their significant serum stability, prolonged storage when frozen, and simplified GMP handling [13].

Some success has been achieved in the use of DEXs for tumor treatment. Treatment of tumor-bearing mice with DEXs derived from DCs loaded with tumor antigens (in the form of proteins, peptides, lysates, or tumor RNA) was shown to activate antitumor cytotoxic T lymphocytes (CTLs) and NK cells, prolong survival of diseased animals, significantly retard tumor growth, and protect mice from tumor development in melanoma, hepatocellular carcinoma, glioma, breast, and cervical cancer mouse models (see our extensive Review [14]). Antitumor DEX-based vaccines were investigated in Phase I and II clinical trials and demonstrated the feasibility and safety of this approach and the ability of vesicular vaccines to activate antitumor T- and NK-cellular immune responses [15,16,17].

Commonly, DEXs are isolated from the conditioned medium of tumor antigen-loaded DCs via sequential ultracentrifugation [14]. Note that several rounds of ultracentrifugation during the process of DEX isolation inevitably resulted in a decreased yield of DEXs [18]. According to our unpublished data, the mean yield of DEXs derived from mouse DCs (bone marrow-derived DCs or tsDC cell line (ECACC 01081609)) was 10 µg per 10 mL conditioned medium, which was comparable with the yields of extracellular vesicles isolated from mesenchymal stem cells [19]. Because the average therapeutic single dose of DEXs is quite high (40–50 µg per mice [20,21,22] and 250 µg (50–2400 µg) per patient [17]), multiple vaccinations are usually used in therapeutic schedules, and additional spend of DEXs in phenotypic and immunological characterization is needed, it is obvious that the isolation of the appropriate amount of DEXs is a difficult challenge. Therefore, alternative methods that provide large-scale production of artificial membrane vesicles (MVs) retaining many characteristics of DEXs are essential. Among the different approaches, the production of MVs by cell treatment with cytochalasin B seems to be the most promising technique.

Cytochalasin B is a cell-permeable fungal toxin that shortens actin filaments by blocking monomer addition at the fast-growing (“barbed”) end of these polymers [23], which results in a reduction of cell membrane rigidity and subsequent release of nanosized cytochalasin B-induced membrane vesicles (CIMVs). This method has been intensively used for CIMV preparation from human mesenchymal stem cells [19,24,25,26], prostate cancer [27], and neuroblastoma [28] cells. CIMVs have been categorized among the classes of MVs or so-called ectosomes, because they originate from direct outward budding from the cell plasma membrane. In turn, CIMVs derived from mature DCs are of special interest due to both the complex of immunostimulating molecules (MHC class II, CD80, CD86) on the surface of the vesicular membrane and the contents of the vesicles (mRNA, miRNA, cytokines). Such “immunocompetent” vesicles can be used as an alternative to DC- and DEX-based vaccines for priming antigen-specific T-lymphocytes and triggering an antitumor response [29,30].

Here, we studied the antitumor potential of cell-free vaccines based on the cytochalasin B-induced membrane vesicles (CIMVs) obtained from DCs loaded with total tumor-derived RNA on a murine model of RLS_40_ lymphosarcoma [31]. We compared the antitumor effects of CIMVs-based vaccines with liposome-based and DC-based vaccines. Tumor growth rate, metastases development, pathomorphological changes in the organs of the immune system, and immunological characteristics of activated antitumor responses (namely, stimulatory/inhibitory immune checkpoints expression and T helper immune response polarization) were assessed.

## 2. Materials and Methods

### 2.1. Liposomes

Cationic spermine/cholesterol-based liposomes L and mannosylated liposomes ML were synthesized as described previously [32,33,34,35]. Liposomes L were composed of the cationic lipid 2X3 (1,26-Bis(cholest-5-en-3β-yloxycarbonylamino)-7,11,16,20-tetraazahexacosan tetrahydrochloride) and DOPE (Avanti Polar Lipids, Alabaster, AL, USA) at a 1:2 molar ratio. Mannosylated liposomes ML were composed of 2X3, DOPE, and mannosylated lipid conjugate (3-[6-(α-D-mannopyranosyloxy)hexyl]amino-4-{6-[*rac*-2,3-di(tetradecyloxy)prop-1-yl oxycarbonylamino]hexyl}aminocyclobut-3-en-1,2-dione) at a 3:6:1 molar ratio.

### 2.2. Cells

B16-F10 murine melanoma cells (hereinafter, B16) were obtained from the Russian Cell Culture Collection (Institute of Cytology, RAS, St. Petersburg, Russia). B16 cells were grown in DMEM medium supplemented with 10% FBS (HyClone, GE Healthcare, Chicago, IL, USA) and 1% antibiotic–antimycotic solution (penicillin (10,000 IU/mL), streptomycin (10 mg/mL), and amphotericin B (25 μg/mL)) (MP Biomedicals, Santa Ana, CA, USA) (hereinafter, complete DMEM) in a humidified atmosphere containing 5% CO_2_ at 37 °C (hereinafter, standard conditions (SC)) and were regularly passaged to maintain the exponential growth.

RLS_40_ drug-resistant murine lymphosarcoma was previously developed from lymphosarcoma LS, susceptible to chemotherapy, in our lab [31]. The RLS_40_ model corresponds to the chemotherapy-resistant tumors of patients who received several courses of chemotherapy [31] and is useful for testing immunotherapy approaches simulating clinical trial conditions in patients eligible for study that already have chemotherapy-resistant cancer cells. RLS_40_ were cultured in complete IMDM medium supplemented with 40 nM vinblastine under SC. Before tumor transplantation, RLS_40_ cells were intraperitoneally (i/p) injected into CBA/LacSto mice (2 × 10^5^ cells per animal) to generate ascites. Ascitic cells were collected and washed with PBS, and RLS_40_ cells were isolated by Histopaque-1083 (Sigma Aldrich, St. Louis, MO, USA) density gradient centrifugation, followed by grafting intramuscularly (i/m) in CBA/LacSto mice (see Methods below).

Mouse dendritic tsDC cells (ECACC 01081609) were purchased from the European Collection of Authenticated Cell Cultures and cultured in IMDM medium supplemented with 5% FBS, 50 µM 2-mercaptoethanol (Sigma Aldrich, USA), 2 mM Glutamax (Thermo Fisher Scientific, Waltham, MA, USA), and 1% antibiotic-antimycotic solution at 33 °C, 5% CO_2_.

Primary immature dendritic cells (imDCs) were obtained from bone marrow DC progenitors of C57Bl/6 mice by Histopaque-1083 (Sigma-Aldrich, USA) density gradient centrifugation, followed by cultivation in IMDM medium containing 10% FBS and 1% antibiotic–antimycotic solution supplemented with 50 ng/mL rmGM–CSF (Sino Biological, Beijing, China) and 50 ng/mL rmIL-4 (Sino Biological, Beijing, China) for 6 days in a humidified atmosphere under SC.

### 2.3. Mice

CBA/LacSto and C57Bl/6 mice of 10- to 12-weeks of age were purchased from the Vivarium of the Institute of Chemical Biology and Fundamental Medicine SB RAS (Novosibirsk, Russia). All animal procedures were carried out in accordance with the recommendations for the proper use and care of laboratory animals (ECC Directive 2010/63/EU). The experimental protocols were approved by the Committee on the Ethics of Animal Experiments at the Institute of Cytology and Genetics SB RAS (Novosibirsk, Russia) (protocol No. 52 from 23 May 2019).

### 2.4. Preparation of Vaccines

#### 2.4.1. Dendritic Cell-Based Vaccines

Total tumor RNA was isolated from murine melanoma B16 or RLS_40_ lymphosarcoma cells (hereinafter, RNA-B16 or RNA-RLS_40_, respectively) with TRIzol Reagent (Invitrogen, Carlsbad, CA, USA) in accordance with the manufacturer’s recommendations. Immature bone-marrow-derived DCs or tsDC cells were transfected with RNA-B16 or RNA-RLS_40_, respectively, by using L or ML liposomes as previously described [33,36] with minor modifications. In brief, complexes of liposomes with RNA (lipoplexes) were formed at N/P (nitrogen-to-phosphate) ratio 4/1 in Opti-MEM medium (Thermo Fisher Scientific, Waltham, MA, USA) at room temperature for 20 min. Then, lipoplexes in serum-free IMDM were added to DCs pre-seeded in 24-well plates or 25 cm^2^ flasks at a rate of 5 µg RNA per 5 × 10^5^ DCs. DCs were incubated in the presence of lipoplexes under SC for 4 h, after incubation, the medium was changed by fresh IMDM medium supplemented with 10% FBS, and cells were further incubated for 20 h under SC. Transfection efficiency of L and ML liposomes was 40–50% of DCs [33]. DC-based vaccines were administered subcutaneously (s/c) at the crest area in the interscapular region at a dose of 10^5^ cells/mouse in 0.2 mL of Opti-MEM. Each time, only freshly prepared DCs were used as DC vaccines.

#### 2.4.2. Liposome-Based Vaccines

Liposome-based vaccines are lipoplexes spontaneously formed in Opti-MEM medium after total tumor-derived RNA is mixed with L or ML liposome solution at N/P ratio 4/1 (for details see [33]). Liposome-based vaccines were administered s/c in the interscapular region at a dose of 5 µg RNA in the form of lipoplexes per mouse in 0.2 mL of Opti-MEM.

#### 2.4.3. Cytochalasin B-Induced Membrane Vesicle (CIMV)-Based Vaccines

CIMVs were prepared as previously described [37] with some modifications. Briefly, immature bone marrow-derived DCs or tsDCs were transfected with RNA-B16 or RNA-RLS_40_ precomplexed with cationic liposomes L or ML at N/P ratio 4/1 as described above. Transfected DCs were washed with serum-free media, resuspended in serum-free IMDM supplemented with 10 µg/mL cytochalasin B (AppliChem GmbH, Darmstadt, Germany) and incubated for 30 min under SC. After that, cells were vigorously vortexed for 30 s. CIMVs were collected by sequential centrifugation at 100× *g* (10 min, 4 °C), 600 g (20 min, 4 °C), and 15,000× *g* (30 min, 4 °C). The resulting pellet was washed with PBS (MP Biomedicals, Santa Ana, CA, USA) and centrifuged at 15,000× *g* (30 min, 4 °C). CIMVs were resuspended in 0.1 mL of PBS and stored at −80 °C until use. CIMV concentration was measured on the basis of the total protein concentration with Qubit Protein Assay kit (Thermo Fisher Scientific, Waltham, MA, USA ) [19]. CIMV-based vaccines were administered s/c at the crest area in the interscapular region at a dose of 15 µg of CIMVs (total protein counts) per mouse in 0.2 mL of Opti-MEM.

### 2.5. Transmission Electron Microscopy of CIMVs

Transmission electron microscopy (TEM) of CIMVs was performed at the Center for Microscopy of Biological Subjects (Institute of Cytology and Genetics SB RAS, Novosibirsk, Russia) using a JEM1400 microscope (80 kV, JEOL, Tokyo, Japan). Negative contrast staining of CIMVs with uranyl acetate and subsequent procedures were performed as previously described [19].

### 2.6. Flow Cytometry Analysis of CIMVs

The CIMV surface proteins were analyzed after the absorption of CIMVs on aldehyde/sulfate latex beads (Invitrogen, USA) followed by blocking with 0.5 M glycine as described previously [19,38]. Latex beads with immobilized CIMVs were stained with monoclonal antibodies for 1 h at 4 °C. Subsequently, the latex beads were washed twice with 300 µL PBS and analyzed by flow cytometry.

Three DC-specific markers involved in the formation of immunological synapse and activation of antitumor T cells were checked on the surface of DC-derived CIMVs, namely, MHC II (FITC; Sony Biotechnology, Tokyo, Japan), CD80 (APC; Sony Biotechnology, Tokyo, Japan), and CD83 (PE; BD Biosciences, San Jose, CA, USA). Flow cytometry measurements were performed using a NovoCyte 3000 flow cytometer (ACEA Biosciences, San Diego, CA, USA) using a 640 nm Ex wavelength for antibodies conjugated to APC (Em 675 ± 30 nm) and 488 nm for antibodies conjugated to FITC or PE (Em 530 ± 30 nm or 572 ± 28 nm, respectively). A forward scatter height versus forward scatter area density plot (linear scales) was used to exclude latex bead doublets. At least 30,000 events were analyzed in each experiment. All experiments were run in triplicate for statistical analysis. Data were processed with FlowJo Software (Version 10.5.3, Tree Star Inc., Ashland, OR, USA).

### 2.7. Evaluation of Antitumor CTL Response Activated by the Vaccines

Healthy male C57Bl/6 mice were divided into four groups and s/c administered in the interscapular region as follows: group 1—saline buffer (control); group 2—bone-marrow-derived DCs transfected with ML/RNA-B16 complexes (10^5^ cells per mouse); group 3—ML/RNA-B16 lipoplexes (5 µg of RNA per mouse); group 4—CIMVs prepared using bone-marrow-derived DCs transfected with ML/RNA-B16 complexes (15 µg of vesicles (total protein counts) per mouse). Mice were sacrificed 7 days post injection, and splenocytes were isolated by density gradient centrifugation (Histopaque-1083). Splenocytes were seeded in complete IMDM medium (1 × 10^6^ cells/mL) supplemented with 20 ng/mL of rmIL-2 (Invitrogen, USA) and melanoma B16 lysate (50 µg total protein counts per mL) and incubated under SC for 6 days to restimulate CTLs. Activated splenocytes (effector cells) were co-cultured with B16 melanoma cells (target cells) at effector-to-target ratio 20:1 in 16-well E-plates (ACEA Biosciences, USA) to estimate the cytotoxic response. Tumor cell viability was monitored for 70 h using the xCELLigence system (ACEA Biosciences, USA).

### 2.8. Tumor Transplantation and Design of Animal Experiments

Female CBA/LacSto mice of 10 to 12 weeks of age were used. Solid tumors of RLS_40_ lymphosarcoma were induced by intramuscular (i/m) injection of tumor cells (10^6^) in 0.1 mL of saline buffer into the right thighs of mice. On day 7 after tumor transplantation, mice were randomized, assigned to one of the experimental groups, and s/c vaccinated with DC-, liposome-, or CIMV-based vaccines at the crest area in the interscapular region once (denoted as (×1), on day 7) or twice (denoted as (×2), on days 7 and 14). DC-based vaccines were 10^5^ of DCs transfected with RNA-RLS_40_/L or RNA-RLS_40_/ML lipoplexes; liposome-based vaccines were 5 µg of RNA-RLS_40_/L or RNA-RLS_40_/ML lipoplexes; CIMV-based vaccines were 15 µg of CIMV prepared from DCs transfected with either RNA-RLS_40_/L or RNA-RLS_40_/ML lipoplexes.

The experimental groups (n = 10 in control and n = 5 in each experimental group) were as follows: (1) control, received Opti-MEM; (2) DC/L(×1), single vaccination with DCs loaded with RNA-RLS_40_/L (d7); (3) DC/L(×2), double vaccination with DCs loaded with RNA-RLS_40_/L (d7, d14); (4) DC/ML(×1), single vaccination with DCs loaded with RNA-RLS_40_/ML (d7); (5) DC/ML(×2), double vaccination DCs loaded with RNA-RLS_40_/ML (d7, d14); (6) L(×1), single vaccination with RNA-RLS_40_/L (d7); (7) L(×2), double vaccination with RNA-RLS_40_/L (d7, d14); (8) ML(×1), single vaccination with RNA-RLS_40_/ML (d7); (9) ML(×2), double vaccination with RNA-RLS_40_/ML (d7, d14); (10) CIMV/L(×1), single vaccination with CIMV prepared from DCs loaded with RNA-RLS_40_/L (d7); (11) CIMV/L(×2), double vaccination with CIMV prepared from DCs loaded with RNA-RLS_40_/L (d7, d14); (12) CIMV/ML(×1), single vaccination with CIMV prepared from DCs loaded with RNA-RLS_40_/ML (d7); (13) CIMV/ML(×2), double vaccination with CIMV prepared from DCs loaded with RNA-RLS_40_/ML (d7, d14).

Starting on day 9 of the experiment, tumor size was determined every 3–4 days by caliper measurements in three perpendicular dimensions. Tumor volumes were calculated as V = (π/6 × length × width × height).

On day 21 of the experiment, the mice were sacrificed. Tumor weight was determined using the equation *W*_tumor_ = *W*_right leg_ − *W*_left leg_, where *W*_right leg_ is the weight of the right leg with tumor, *W*_left leg_ is the weight of the left leg without tumor. The animal experiment was performed twice.

Tumor tissues and organs (liver, spleen, thymus) were collected for histology, immunohistochemistry, and morphometry. Peripheral blood was collected from the retroorbital sinus for serum preparation and for blood biochemistry assessment. Total RNA was isolated from splenocytes with TRIzol Reagent according to manufacturer protocol. RNA was used to analyze the expression levels of immune checkpoints and master regulators of T cell differentiation.

### 2.9. Histology and Immunohistochemistry

Tissue specimens (tumor nodes, liver, spleen, and thymus) were fixed in 10% neutral buffered formalin (BioVitrum, Moscow, Russia), dehydrated in ascending ethanols and xylols, and embedded in HISTOMIX paraffin (BioVitrum, Russia). The paraffin sections (5 μm) were sliced on a Microm HM 355S microtome (Thermo Fisher Scientific, Waltham, MA, USA) and stained with hematoxylin and eosin. For the immunohistochemical study, the tumor sections were deparaffinized and rehydrated. Antigen retrieval was carried out after exposure in a microwave oven at 700 W. The samples were incubated with anti-caspase-3 (ab2302, Abcam, Cambridge, UK) primary antibodies according to the manufacturer’s protocol. Then, the sections were incubated with secondary horseradish peroxidase (HPR)-conjugated antibodies (Spring Bioscience detection system, Spring Bioscience, Pleasanton, CA USA), exposed to the 3,3′-diaminobenzidine (DAB) substrate, and stained with Mayer’s hematoxylin. All images were examined and scanned using an Axiostar Plus microscope equipped with an Axiocam MRc5 digital camera (Zeiss, Munich, Germany) at magnifications of ×100 and ×400. Five to ten random fields were studied in each specimen depending on specimen size, forming 25–50 random fields for each experimental group of mice in total. In the control group containing 10 mice, 50–100 random fields were analyzed.

The percentages of the internal metastases areas in the liver were determined relative to the total area of liver sections using Adobe Photoshop. Inhibition of metastases development was assessed using the metastasis inhibition index (MII), calculated as MII = ((mean metastasis area_control_ − mean metastasis area_experiment_)/mean metastasis area_control_) × 100%. MII in the tumor-bearing mice without treatment (control) was taken as 0%, and MII, reflecting the absence of metastases, was taken as 100%.

Morphometric analysis of tumor, spleen, thymus, and liver sections was performed by point counting, using a morphometric grid with 100 testing points in a testing area equal to 3.2 × 10^6^ μm^2^. Morphometric analysis of tumor tissue included evaluation of the volume densities (Vv, %) of unchanged tumor tissue, inflammatory infiltration, necrosis, and numerical densities (Nv) of mitoses and caspase-3-positive cells. Morphometric analysis of the spleen included evaluation of the volume densities (Vv, %) of red pulp, white pulp, and the diameter of lymphoid follicles (µm). Morphometric analysis of the thymus included evaluation of the volume densities (Vv, %) of the cortex and medulla with subsequent calculation of cortex/medulla index. Morphometric analysis of the liver included evaluation of the volume densities (Vv, %) of normal liver tissue, dystrophy, necrosis, and numerical density (Nv) of binuclear hepatocytes.

The volume density (Vv, %) of the histological structure under study indicates the volume fraction of tissue occupied by this compartment, is determined from the testing points lying over this structure, and is calculated using the following formula: Vv = (P_structure_/P_test_) × 100%, where P_structure_ denotes the number of points over the structure and P_test_ denotes the total number of test points, 100 in this case. The numerical density (Nv) of the histological structure under study indicates the number of particles in the unit of tissue volume evaluated as the number of particles in the square unit, 3.2 × 10^6^ μm^2^ in this case.

### 2.10. Blood Biochemistry

Biochemical parameters of blood serum of experimental animals were estimated using an HTI BioChem FC-200 Auto Chemistry Analyzer (HTI, Farmington, MI, USA). The serum levels of alanine aminotransferase, ALT (HT-A206-120, HTI, USA); aspartate aminotransferase, AST (HT-A109-120, HTI, USA); alkaline phosphatase, ALK (HT-A205-120, HTI, USA); and total protein (HT-T251-125, HTI, USA) were evaluated.

### 2.11. Immune Checkpoints and Master Regulators Analysis

To analyze the type of activated T-helper immune response and the expression levels of negative and positive immune checkpoints after treatment of RLS_40_-bearing mice with different vaccines, a detection system was developed on the basis of qPCR and fluorescent probes. To design primers and probes, we utilized the RealTime PCR Tool (https://eu.idtdna.com/scitools/Applications/RealTimePCR/, accessed on 15 July 2019). The secondary structure of the primers/probes and the self- and heterodimerization tendencies of each primer/probe set were predicted using the OligoAnalyzer Tool (https://eu.idtdna.com/calc/analyzer, accessed on 15 July 2019). Primers and probes were synthesized in the Laboratory of Biomedical Chemistry of ICBFM SB RAS (Novosibirsk, Russia).

Total RNA was isolated from the splenocytes of experimental animals using TRIzol Reagent in accordance with manufacturer recommendations. Coding DNA (cDNA) was synthesized from total RNA in 20 µL of reaction mixture containing 5 µg total RNA, 4 µL 5× RT buffer (RevertAid, Termo Fisher, Waltham, MA, USA), 200 U RevertAid (TermoFisher, USA), 1 mM dNTP mixture, and 100 µM random hexaprimer. Reverse transcription was performed at 25 °C for 10 min followed by incubation at 42 °C for 60 min. Finally, the reverse transcriptase reaction was terminated at 70 °C for 10 min.

PCR was carried out in a total volume of 25 µL using 10× F2 buffer (65 mM Tris-HCl, pH = 8.9; 16 mM (NH_4_)_2_SO_4_, 1.5 mM MgCl_2_; 0.05% Tween 20; 10 mM 2-mercaptoethanol) (BIOSSET Ltd., Novosibirsk, Russia); dNTP mixture (0.4 mM of each dNTP) (Promega, Madison, WI, USA); 250 nM TaqAT (Hot start Taq polymerase 40 e.a./mL) (ThermoScientific, USA); 0.4 µM of each forward and reverse primers to β-actin and β-actin-specific ROX-labeled probe; 1× PCR additive (5M Betaine) (Sigma-Aldrich, cat. no. B0300-5VL, USA); and 0.4 µM of each forward and reverse gene-specific primers and FAM-labeled probes (Table 1). Amplification was performed as follows: (1) 94 °C, 2 min; (2) 94 °C, 10 s; 60 °C, 30 s (45 cycles); (3) 10 °C, 2 min. The relative level of gene expression was determined by the CFX96^TM^ Real-Time system (Bio-Rad, Hercules, CA, USA) and normalized to the level of β-actin. The obtained PCR data were analyzed using CFX Maestro Software Version 1.0 (Bio-Rad, Hercules, CA, USA).

### 2.12. Statistical Analysis

The data were statistically processed using one-way ANOVA with the Tukey or Fisher post hoc tests; a *p* value of ≤0.05 was considered to indicate a significant difference.

## 3. Results

### 3.1. Preparation and Characterization of Vaccines

In this study, three types of vaccines were investigated. Cellular DC-based vaccines were prepared from mouse bone marrow-derived DCs or tsDCs transfected with tumor cell-derived RNA pre-complexed with either liposomes L or ML as previously described [33,39]. Lipoplexes were pre-formed at a nitrogen-to-phosphate (N/P) ratio of 4/1 because, at this ratio, the optimal combination of transfection efficiency and DC targeting was observed [33,39]. Liposome-based vaccines were lipoplexes of the same liposomes (L or ML) with tumor RNA prepared at N/P ratio 4/1. Vesicular vaccines were cytochalasin B-induced membrane vesicles (CIMVs) prepared from DCs loaded with total tumor-derived RNA pre-complexed with liposomes L or ML. Altogether, six different vaccines were used, and each vaccine was administered once (×1) or twice (×2) (Table 2).

The mean yield of DC-derived CIMVs (by total protein count) was 100–150 µg per 10^7^ tsDCs, more than 10-fold higher compared with the yields of natural extracellular vesicles of DCs isolated from the conditioned medium [37].

Transmission electron microscopy revealed that DC-derived CIMVs had the same characteristic spherical structures, most of which had size ≤ 150 nm (Figure 1A), as our previous data obtained using dynamic light scattering, nanoparticle tracking analysis, and TEM techniques [37]. As previously shown, lipoplexes formed by tumor RNA and liposomes L or ML at N/P = 4/1 were rather small (Ø 80–150 nm) and characterized by a polydispersity index (PDI) of 0.2, showing the formation of uniform particles [33]. DC-derived CIMVs were also characterized by small diameters (50–150 nm) and low PDI of 0.47, which indicated the formation of small and rather uniform particles similar to the lipoplexes [37]. Additionally, DC-derived CIMVs were demonstrated to express on their surface the DC-specific proteins MHC II, CD80, and CD83, essential for the formation of immunological synapses with T-lymphocytes and subsequent activation of the antitumor immune response (Figure 1B).

The main challenge in the application of antitumor vaccines of different origins, namely, cellular, liposome-, and vesicle-based ones, in one experiment is the correlation of the dosages between different vaccine types. We adjusted the doses of vaccines based on the number of DCs or amount of tumor RNA used for vaccine preparation per mouse. DC-based vaccines were injected at a dose of 10^5^ cells (1 µg tumor RNA) per mouse, in accordance with our previous data [32,33,39] and other studies [40,41]. This dose of DCs was sufficient to activate the antitumor immune response and inhibit tumor growth and metastasis in tumor-bearing mice [32,33,39,40,41]. Liposomal vaccines containing mRNA are commonly administered to mice at a dose of 10–30 µg RNA per mouse [42,43,44,45]. Here, we used a lower dose of liposomal vaccine 5 µg RNA per mouse because it was previously shown that this dosage efficiently activates antitumor CTLs in vivo in murine melanoma B16 model [36]. Note that the amount of RNA per single dose in liposomal and DC-based vaccines differs five-fold; it is supposed that this is the minimal amount needed to overcome the loss of lipoplexes due to non-targeted delivery in vivo after s/c administration.

We consider CIMVs as an alternative to natural extracellular vesicles (exosomes). Hence, the dosage of DC-derived CIMVs was calculated while considering both the amount of exosomes used worldwide for antitumor vaccination and the number of DCs used to prepare CIMVs to keep the dosage of DC- and CIMVs-based vaccines as close as possible. DC-derived extracellular vesicles are commonly administered to experimental animals at a dose of 40–50 µg (by total protein content) per mouse, although some researchers have used higher (100 µg per mouse [46]) or significantly lower (1–3 µg per mouse [47]) dosages. In our work, a dose of 15 µg of CIMVs (by total protein content) per mouse was used that met both criteria listed above. On the one hand, the dose of CIMVs is comparable to that of exosomes; on the other, a single dose of CIMVs was prepared from 10^6^ of DCs that was only 10-fold higher than the dose of the DC-based vaccine where the excess of CIMVs was required to compensate for the loss of CIMVs due to non-productive interactions with non-immunological cells and tissues upon s/c administration. This concentration of CIMVs was previously used in animal experiments by Gomzikova et al. [24].

### 3.2. DC-Derived CIMVs Efficiently Primed Antitumor Cytotoxic T-Cells In Vivo

The potential of DC-, liposome-, and CIMV-based vaccines to activate antitumor CTLs in vivo was assessed. Because RLS_40_ lymphosarcoma is a tumor that is maintained mainly in ascites and poorly handled in culture, we decided to perform experiments in a melanoma B16 model. For this purpose, splenocytes were isolated from healthy C57Bl/6 mice 7 days after s/c vaccination with the respective vaccine and restimulated for 6 days ex vivo with the lysate of B16 cells. Restimulated splenocytes (effector cells) were co-cultured with B16 cells (target cells) at effector-to-target cell ratio 20/1, and cytotoxic activity of splenocytes was monitored in real-time mode using the xCELLigence system (ACEA Biosciences, USA) (Figure 2). It was shown that all tested vaccines stimulated antitumor CTLs that efficiently lysed tumor cells, showing up to a 30-fold decrease in the cell index of melanoma B16 cells compared with the control (Figure 2). Among tested vaccines, the liposome-based one was a slightly less efficient activator of antitumor CTLs compared with DC- and CIMV-based vaccines. Thus, DC-, liposome-, and CIMV-based vaccines under the study were demonstrated to efficiently activate antitumor CTLs in vivo.

### 3.3. Effect of Vaccination on RLS_40_ Lymphosarcoma Growth and Metastasis Development In Vivo

The experimental setup for investigating the antitumor potential of the designed vaccines is depicted in Figure 3A. RLS_40_ cells (1 × 10^6^) were implanted i/m into the right thigh of CBA mice. After that, on day 7, animals were randomized, assigned to one of the experimental groups (n = 10), and vaccinated s/c with DC-, liposome-, or CIMV-based vaccines. Five animals from each group under study (see Table 2 for details) received a second dose of vaccine on day 14 (Figure 3A). Altogether, 12 experimental and one control (non-treated) groups were included in the experiment. Tumor size was monitored every 3–4 days, followed by calculation of tumor volumes (Figure 3B). Mice were euthanized on day 21 of tumor growth, followed by the determination of tumor weight as well as collection of tumor nodes and organs for subsequent histological analysis (Figure 3C).

It is obvious from Figure 3B that all types of tested vaccines caused 4–5-fold inhibition of the growth of primary tumor nodes compared with the control group. However, differences in the tumor volume between the experimental groups were statistically insignificant. For example, under a single vaccination schedule (Figure 3B, upper left panel) DC-based vaccine DC/ML and both non-cellular vaccines (lipoplexes RNA-RLS_40_/ML and CIMVs derived from loaded DCs; CIMV/ML) exhibited similar inhibition of tumor growth at the end of the experiment. Interestingly, some faint differences in the tumor suppression rate appeared between the vaccines on the 15th day after tumor transplantation: DC/ML looked slightly less efficient than ML and CIMV/ML. In the case of double vaccination with the same vaccines, CIMV/ML again seems somewhat more efficient, since no tumor growth was observed during the entire period of observation, while in the case of DC/ML and ML vaccines, some tumor growth was observed starting on day 15 of the experiment (Figure 3B). Interestingly, preparation of antitumor vaccines using DC-targeted mannosylated liposomes ML provides almost no advantages over the application of non-targeted cationic liposomes L (Figure 3B). We previously observed this fact, and it can be explained by the superior transfection efficiency of liposomes L, especially in vitro [32,33]. However, in vivo, single and double vaccination of mice with vaccine L led to vigorous inhibition of tumor growth, which was even more visible in the case of single vaccination (Figure 3B, low left panel); in the case of DC/L and CIMV/L vaccines, visible growth of tumor was observed starting on day 12 of observation, while in the group L vaccine, the tumor growth was not observed. One of the possible explanations is that, being positively charged, L vaccine caused non-specific activation of the immune system that resulted in tumor growth inhibition. Similar effects of cationic liposomes on the immune system are well-documented [48,49].

Thus, two schedules of vaccination of tumor-bearing mice were used in the study, including (1) single vaccination on day 7 after the tumor transplantation and (2) double vaccination on day 7 followed by booster vaccination on day 14. Both regimens of vaccination were demonstrated to be efficient and resulted in significant retardation of RLS_40_ growth compared with the control group (Figure 3B). Some advantages of double vaccination were observed only for the groups vaccinated with DC/L vaccine, in which double vaccination was much more effective compared with a single dose. Similarly, double vaccination with CIMV/ML was also more effective than a single dose.

It was shown that tumor weight of vaccinated mice decreased 4–8-fold in all groups compared to the control (Figure 3C). These data correlated well with the results of vaccine-induced retardation of tumor growth (Figure 3B,C). Furthermore, statistically significant differences between the tumor weights in different groups were observed (Figure 3C). The more pronounced effect of vaccination on tumor weight was observed in the groups double vaccinated with DC/L, CIMV/ML, and once vaccinated with CIMV/L. In these groups, no tumor growth, as well as minimal (as low as 0.1–0.2 g) tumor weight, were registered. In other groups, tumor weights decreased after vaccination but not significantly.

Thus, DC-, liposome-, and CIMV-based vaccines significantly retarded the growth of the primary tumor node. However, in the total application of the DC-targeted mannosylated liposomes, ML instead of L and administration of booster vaccination did not significantly improve the antitumor effect of the vaccines.

Given the very efficient inhibition of RLS_40_ tumor growth in vaccinated groups of mice, we further analyzed the effect of vaccination on the development of internal metastasis in the liver: the target organ of lymphosarcoma RLS_40_. Histologically, liver metastases of the lymphosarcoma RLS_40_ were represented predominantly by rounded foci with unclear boundaries consisting of the large monomorphic atypical lymphoid cells, which were comparable with the cells forming the primary tumor node. The histological structure of the primary tumor node of RLS_40_ lymphosarcoma has been previously described in detail [50,51]. The areas occupied by metastases were determined relative to the total area of the liver section and expressed with the Metastases Inhibition Index (MII) (for details, see Section 2). As depicted in Appendix A, vaccine administration resulted in a significant inhibition of metastasis development: all types of vaccines efficiently suppressed the development of RLS_40_ internal metastases in the liver with an MII of 73.8–99.2% (Appendix A). The anti-metastatic effect of DC/L and DC/ML vaccines ranged from 73.8% to 94% and tended to increase with twofold immunization (Appendix A). After administration of L or ML lipoplexes loaded with RNA-RLS_40_, MII reached 93.1% for L and 99.2% for ML, thus showing that ML vaccine was somewhat more effective; however, no statistically significant differences between single and twofold immunizations were found (Appendix A). After administration of CIMV obtained from L/RNA-RLS_40_- or ML/RNA-RLS_40_-loaded DCs, MII reached 87.6–99% and tended toward a slight decrease with double immunization (98–99% for single and 91.8–87.6% for double immunization) (Appendix A). Thus, there were no statistically significant differences concerning the antimetastatic activity between the groups; however, cell-free vaccines (L, ML, CIMV/L, and CIMV/ML) were more effective than DC-based vaccines, and booster immunization was more effective in the case of DCs and less effective in the case of CIMV-based vaccines. The use of DC-targeted or non-targeted liposomes for delivery of tumor-derived RNA to DCs did not affect the antimetastatic activity of different types of vaccines.

A histological study of RLS_40_ lymphosarcoma tissue revealed that vaccines caused changes in the histological characteristics of the primary tumor node manifested in the increase of foci of necrosis in the central part of the tumor node and inflammatory infiltration located on the border between the unchanged and necrotic tissue and was represented mainly by neutrophils with slight lymphocyte/macrophage admixture. Morphometric analysis revealed that the volume density of necrotic changes in the control RLS_40_ tumors was 4.8 ± 0.5% (Figure 4, Appendix A). The administration of vaccines led to a slight increase in necrotic decay; however, this parameter was two-fold increased in the groups that received DC/ML(×2) and ML(×2) vaccines: the volume density of necrosis was 9 ± 1.5% and 10.2 ± 4.1%, respectively (Figure 4, Appendix A). The volume density of inflammatory infiltration in the control RLS_40_ tumors was 6.2 ± 1.5%, and vaccination with DC/L(×2), DC/ML(×2) and CIMV/ML(×2) led to 3.5-, 2.2-, and 2-fold increase of this parameter compared to the control group (Figure 4, Appendix A).

RLS_40_ tissue is characterized by high mitotic activity: the numerical density of tumor cells in the state of mitosis was 5.6 in the test area (Figure 4, Table 3), while vaccination with DC- and CIMV-based vaccines reduced mitotic activity in tumor tissue: the number of mitoses decreased 2.8-, 3.2-, 2.2-, and 2.7-fold after double immunization with DC/L, DC/ML, CIMV/L, and CIMV/ML, respectively, as compared with the control group (Figure 4, Table 3). RLS_40_ tissue was initially characterized by a certain level of spontaneous apoptosis: the numerical density of caspase-3-positive tumor cells was 1.4 ± 0.1 per test area (Figure 4, Table 3). Double administration of DC- and MV-based vaccines led to 4.7- (DC/L), 3.4- (DC/ML), 4.5- (CIMV/L), and 4.1-fold (CIMV/ML) increases in the number of caspase-3-positive cells compared with the control group (Figure 4, Table 3), however, single immunization with these types of vaccines did not induce apoptosis in tumor tissue (Table 3). As for liposome-based vaccines, both single and double vaccinations with these vaccines were ineffective with respect to mitotic activity and apoptosis induction in tumor tissue (Table 3).

### 3.4. Liver Toxicity of Vaccines in RLS_40_-Bearing Mice

Cationic liposomes accumulate and undergo biotransformation in the liver, causing toxic effects. To assess the damaging potential of vaccines under the study, morphological changes in the liver tissue and biochemical parameters of the peripheral blood, showing hepatic destruction, were assessed.

We demonstrated that the vaccines under study did not induce significant hepatotoxicity. Indeed, the blood biochemistry of mice with RLS_40_ showed that tumor development led to a 1.7-fold increase in the AST level compared with the healthy animals (Appendix A). The values of the other parameters under study (ALT, ALK, and total protein) in tumor-bearing mice did not differ from the healthy level (Appendix A). Single and double vaccination did not increase parameters reflecting the cytolysis of hepatocytes (ALT and AST) compared to the control but led to 1.4–1.9-fold increase in the parameter reflecting the cholestasis (ALK), especially during boosted treatment (Appendix A). Protein synthetic liver function was not affected in any experimental groups (Appendix A).

Liver morphometry of the RLS_40_-bearing mice demonstrated that tumor development was accompanied by the expansion of destructive changes in the liver parenchyma (Appendix A): the volume densities of dystrophy and necrosis were increased by the factors of 1.4 and 4.2, respectively, compared with the healthy mice (Appendix A). Vaccination with DC/L(×2) and L(×2) was the most damaging and was accompanied by an increase in the total destructive changes in the liver by 1.6- and 1.5-fold compared with control and by 3.6- and 3.3-fold compared with healthy mice, respectively (Appendix A). All other types of vaccines did not enhance the destructive effect of tumor growth on the liver parenchyma of tumor-bearing mice (Appendix A).

### 3.5. Immunomodulatory Effect of Different Types of Vaccines in RLS_40_-Bearing Mice

Histologically, the spleen of control RLS_40_-bearing mice had a typical structure and was comparable with the spleen of healthy mice: the white pulp was developed moderately and consisted of lymphoid follicles, most of which were isolated from each other (Figure 5 and Figure 6, Appendix A). The structural organization of the thymus in the control group also did not differ from that of healthy mice and was represented equally by cortex and medulla (Figure 5 and Figure 6, Appendix A). The morphometric study of the spleen and thymus revealed that the tumor progression had no immunomodulatory effects: morphofunctional parameters of the spleen and thymus did not differ from those of healthy animals (Figure 5 and Figure 6, Appendix A).

Both single and double vaccination of tumor-bearing animals activated the immune response, which was expressed in an increase in the volume density of the white pulp and the diameter of lymphoid follicles in the spleen as well as in an increase in the volume density of the thymus cortex and cortex-medulla index in the thymus (Figure 5 and Figure 6, Appendix A). The most efficient immunomodulators were DC- and CIMV-based vaccines, particularly during boosted treatment. Administration of DC/ML (×2) led to a 1.6- and 1.3-fold increase in the volume density of the white pulp and the diameter of the lymphoid follicles of the spleen, respectively, and a 1.4- and 2.3-fold increase in the volume density of the cortex and cortex–medulla index in the thymus, respectively, compared to healthy and untreated tumor-bearing mice (Figure 6, Appendix A). The administration of CIMV/ML (×2) led to a more significant stimulation of the spleen and thymus: the white pulp and lymphoid follicles of the spleen were increased by a factor of 1.7 and 1.4, respectively, and the thymus cortex and cortical–cerebral index were increased by a factor of 1.7 and 11.8, respectively, compared with healthy and control animals (Figure 6, Appendix A). L and ML liposome-based vaccines activated the immune organs to a lesser extent. Thus, double immunization with CIMV-based vaccines demonstrated the most pronounced immunomodulatory effects.

### 3.6. Immunologic Characteristics of Antitumor Immune Responses Caused by Vaccination

Antitumor and antimetastatic activity of the vaccines (Figure 3 and Appendix A) was mediated by stimulation of adaptive immunity, including T lymphocytes, that was corroborated by our data from CTL assays for DC- [33], liposome- [36], and CIMVs-based (Figure 2) vaccines as well as morphological changes in the spleen and thymus of experimental animals revealed by morphometry (Figure 5 and Figure 6, Appendix A). To study the characteristics of antitumor responses activated by the vaccines in detail, analysis of the expression of immune checkpoints (ICs) in immune cells and polarization of T helper responses were performed.

The expression levels of ICs, namely positive (CD27, CD28, CD40L, 4-1BBL) and negative (PD-1, CTLA4, TIGIT, TIM3) regulators of immune response, as well as master regulators of Th cell development (T-bet, GATA3, RORγ, Foxp3), were analyzed in splenocytes of RLS_40_-bearing mice challenged with the vaccines under the study using qPCR (Figure 7).

Among stimulatory ICs, moderate upregulation of CD27 and CD28 expression was observed in the groups once injected with DC-based and liposomal vaccines, whereas double vaccination resulted in downregulation of the expression of these ICs (Figure 7A,B). CIMV-based vaccines tended to slightly upregulate the expression of CD27 (Figure 7A) and reduce the CD28 expression 2–3-fold compared with the control (Figure 7A,B).

Surprisingly, CD40L expression was strongly downregulated in all experimental groups compared with the control (Figure 7C). Generally, CD40L is expressed mostly on activated CD4^+^ T-helpers and, on the one hand, promotes co-stimulatory activity of APC that is crucial for generating CD8^+^ T cell immunity [52]. On the other hand, CD40L is essential for inducing Th2 cells and the subsequent activation of humoral immunity [53,54]. Hence, vaccination-induced downregulation of CD40L may result from decreased activation levels of CD4+ T cells associated with low tumor burden in vaccinated groups. Furthermore, CD40L expression is characterized by fast kinetics and up-regulated within hours after stimuli and declines to a basal level during the next 24 h [52]. However, in this study, CD40L levels were evaluated at the end of the experiment, at least 7 days after the booster vaccine administration. However, a more likely explanation of the drop in CD40L expression levels is that vaccines under the study mediated the switch from the Th2-mediated humoral immune response observed in tumor-bearing animals (control) to Th1 immunity and antitumor CTL development. This statement was confirmed by analyzing the expression levels of Th-specific master regulators (see below).

The expression of 4-1BBL, an antigen-presenting cell-derived immunostimulating checkpoint, was upregulated in splenocytes of mice once challenged with DC- and liposome-based vaccines and with CIMV/L vaccine (Figure 7D), whereas CIMV/ML in contrast increased 4-1BBL expression after double immunization. Further, 4-1BBL expression was up-regulated 1.5–2-fold in the order CIMV/L(×1) ≈ CIMV/L(×2) ≈ CIMV/ML(×2) > DC/L(×1) ≈ DC/ML(×1) ≈ L(×1) ≈ ML(×1).

Regarding negative immune checkpoints, all tested vaccines significantly downregulated the expression of PD-1, one of the most critical inhibitory molecules suppressing T cell-mediated antitumor response (Figure 7E). DC- and liposomal-based vaccines resulted in a two-fold decrease in CTLA4 expression compared with the control, whereas CIMV-based vaccines had no effects on the expression of this factor (Figure 7F). Nevertheless, CIMV-based vaccines efficiently downregulated the expression of another co-inhibitory factor, TIGIT, regardless of the number of vaccine administration (Figure 7G), whereas DC- and liposome-based vaccines efficiently downregulated TIGIT expression only in the case of double vaccination (Figure 7G). Downregulation of TIM3 expression was observed after double vaccination with DC- and liposome-based vaccines and CIMV/ML vaccine. A single administration of any vaccines under study did not affect TIM3 expression (Figure 7H).

Analysis of the expression of master-regulators controlling T-helper differentiation revealed that increase in Tbet expression was observed in all experimental groups that received a single vaccination (Figure 8A) and did not decline to the control level in the groups DC/L, ML, and CIMV/ML after a double vaccination, indicating the activation of the Th1 response. Th2-related transcription factor GATA3 was strongly downregulated in all groups of vaccinated mice (Figure 8B). The expression level of the RORγ master regulator inducing the development of Th17 immune response was shown to be significantly upregulated, by 4–11-fold, compared with the control after both single and double vaccination with all types of tested vaccines, excluding DC/L(×2) and CIMV/L(×2). In all cases, single vaccination was more efficient than double one in stimulation of RORγ expression (Figure 8C). Foxp3 transcription factor is responsible for developing T regulatory cells and its level represents the extent of suppression of T cell response. It was demonstrated that the levels of Foxp3 expression were not increased after vaccination in all groups compared with the control (Figure 8D), indicating the lack of induction of T regulatory cells. It is worth mentioning that up to a 2-fold decrease in Foxp3 expression levels was observed after vaccination in some groups (see Figure 8D).

Thus, we showed that RLS_40_ progression in mice led to the formation of a predominant Th2-mediated pro-humoral immune response that was confirmed by the activation of GATA3 with low levels of T-bet and RORγ and a high level of CD40L immune checkpoints related to B cell development and activation. Vaccination of RLS_40_-bearing mice with DC-, liposome-, or CIMV-based vaccines resulted in switching to cellular immunity by activation of Th1/Th17 cells as well as the induction of positive immune checkpoint 4-1BBL and downregulation of suppressive immune checkpoints in a raw PD-1 >>> TIGIT > CTLA4 > TIM3. All these immunological alterations caused significant retardation of tumor growth and inhibition of metastasis development.

## 4. Discussion

Cell-free antitumor vaccines have many advantages over cellular vaccines, such as prolonged storage without loss of immunotherapeutic activity, insensitivity to the immunosuppressive tumor microenvironment, and potentially better standardization. Therefore, the development of anticancer cell-free vaccines has great potential.

Previously, we developed DC-based vaccines composed of murine BM-DCs loaded with total tumor RNA by using cationic liposomes L or DC-targeted mannosylated liposomes ML that exhibited high antimetastatic activity in a murine melanoma B16-F10 model in vivo [32,33,36]. In the present study, these DC-based vaccines were applied for the treatment of drug-resistant lymphosarcoma RLS_40_ and demonstrated superior antitumor and antimetastatic activity (Figure 3, Appendix A). It is interesting that the change in the route of administration of DC vaccine from intravenous [32,33] to subcutaneous did not affect the antitumor activity of the vaccine.

The main result of this study is the strong antitumor and antimetastatic activity of cell-free liposome- and DC-derived CIMV-based vaccines in a murine RLS_40_ lymphosarcoma model in vivo, with the efficiency being similar to that of DC-based anticancer vaccines.

The antitumor potential of liposomal vaccines, more commonly called mRNA vaccines, is under intensive investigation, especially after the success of mRNA vaccines developed for the prevention of SARS-CoV-2. Antitumor liposomal vaccines are generally composed of cationic liposomes/polymers and mRNAs encoding particular tumor-associated antigens (TAA), such as TRP2 [55,56], MUC1 [57], cytokeratin 19 [58], E7 [56,59], neoantigens [60], etc. or model antigen OVA [43,56,59,61,62]. These mRNA vaccines efficiently delivered TAA-encoding mRNA to DCs and subsequently activated antitumor CD8^+^ and CD4^+^ T-cell immune responses, reduced tumor-associated immunosuppression, and finally significantly decreased tumor growth and metastasis in different murine tumor models [43,55,56,57,58,59,60,61,62]. Our liposomal vaccines were composed of cholesterol/spermine-based cationic liposomes L or DC-targeted mannosylated liposomes ML complexed with total tumor-derived RNA (source of multiple tumor antigens). Their significant potential to deliver RNA in murine DCs ex vivo and activate antitumor CTLs in vivo was demonstrated in our previous studies [32,33,36]. Here, we demonstrated that liposomal vaccines under the study had great antitumor potential in vivo, significantly suppressing RLS_40_ lymphosarcoma growth (Figure 3), inhibiting hepatic metastasis (Appendix A), and activating antitumor immunity (Figure 6, Figure 7, Figure 8, Appendix A). Note that there were no significant differences between the in vivo antitumor activity of the vaccines based on L- and DC-targeted ML liposomes (Figure 3, Appendix A), except for lower hepatotoxicity of ML liposomes (Appendix A). Indeed, in our previous work, we demonstrated an equal efficiency of core L liposomes and their derivative ML liposomes to deliver RNA in murine DCs ex vivo [33], which resulted from the basic superior transfection activity of core L liposomes.

Another type of cell-free vaccine studied in this work use artificial vesicles prepared from loaded DCs using cytoskeleton-disrupting agent cytochalasin B. Being analogous to natural extracellular vesicles, CIMVs represent submicrometer-sized vesicles surrounded by plasma membranes comprising functional cell surface receptors on the outer vesicle surface and cytosolic nucleic acids and proteins in the vesicle interior [63]. Currently, biological activities of membrane vesicles obtained from cytochalasin-B-treated cells have been actively investigated. A significant contribution to this area was made by the laboratory of Prof. Rizvanov A.A. of Kazan Federal University, Russia [24,25,27,28,64,65,66,67]. To date, delivery efficiency, angiogenic, and immunomodulatory properties of CIMVs obtained from human [19,25,37,64,65,66] and mouse [24] mesenchymal stem cells, human neuroblastoma SH-SY5Y [28,67], prostate cancer PC3 [27], and HeLa cells [68] have been investigated. CIMVs were demonstrated to be non-toxic, have no fusion specificity to the cells in vitro, and possess angiogenic activity. Immunomodulatory activities of CIMVs depended on the presence of immune-related molecules on their surface or inside the vesicles. Indeed, CIMVs derived from human mesenchymal stem cells overexpressed IL-2 [25] or TRAIL, while PTEN and IFN-β1 [65] were demonstrated to stimulate the proliferation of T lymphocytes in vitro. Furthermore, CIMVs obtained from HeLa cells transduced with genes encoding tumor-associated antigens significantly activated and maintained the antigen-specific stimulation and expansion of CAR-T lymphocytes [68]. Nevertheless, the immunostimulatory properties of DC-derived CIMVs have been insufficiently studied. Recently, Oshchepkova et al. have demonstrated that DC-derived CIMVs could transfer nucleic acids to the cells without any fusion specificity [37]. In the present study, we demonstrated for the first time the high antitumor and immunostimulatory activity in vivo of CIMVs obtained from murine DCs. We showed that CIMVs prepared from DCs loaded with total tumor RNA can directly prime antitumor CTLs ex vivo (Figure 2) and activate potent antitumor immune response in vivo (Figure 3, Appendix A, Figure 7 and Figure 8), which is similar to or even higher than that of DC-based vaccines.

Now consider common and different features of immune responses activated by different types of the vaccines under the study. It was demonstrated that DC- and CIMV-based vaccines being administered subcutaneously had comparable potential to stimulate highly efficient antitumor CTLs in vivo, whereas liposomal vaccines were weaker CTL inducers (Figure 2). Previously, we have shown that CTLs activated in vivo by intravenously administered liposomal vaccines lysed tumor cells with the same efficiency as DC-based vaccines [36]. Undoubtedly, the route of administration of the vaccines affects their activity. Hence, intravenous inoculation of liposomal vaccines should potentially be preferred over subcutaneous one. Despite different levels of CTL induction, the antitumor efficiencies of the different vaccines were similar: tumor volumes/weights (Figure 3) and the amounts of liver metastasis (Appendix A) were significantly decreased, regardless of the vaccine type. However, different details of immune responses activated by the vaccines and their diverse effects on tumor tissue were observed. Indeed, immunostimulating properties of DC- and CIMV-based vaccines had common patterns, whereas liposomal vaccines were slightly different from them, namely, in weaker activation of lymphoid organs (minimal volume of white pulp in spleen and cortex in thymus especially for L liposomes (Figure 6, Appendix A)) and lower volumes of inflammatory infiltration in tumor tissues (Appendix A). Additionally, a greater number of mitotic tumor cells and a lesser activation of apoptosis in tumor tissue were observed in the case of double vaccination with liposomal vaccines (Table 3). Nevertheless, the changes in the expression levels of immune checkpoints triggered by liposomal vaccines were conversely similar to those of DC-based vaccines, whereas the effects of CIMV-based vaccines differed (Figure 7). It was demonstrated that CIMV-based vaccines more efficiently activated the expression of 4-1BBL and CTLA4 and suppressed the expression of CD28, TIGIT, and TIM3 compared with DC- and liposomal vaccines. Despite these indicated differences, all the vaccines under the study induced Th1/Th17 immune response, with a more polarized Th17 response in the case of CIMV-based vaccines (Figure 8).

Interestingly, the booster vaccination did not improve the overall antitumor efficacy of the vaccines under the study (Figure 3, Appendix A). A similar absence of the advantages of booster DC-based was previously observed [39]. However, some differences in the histological characteristics of tumor nodes and the details of activated immune responses were found between single and double vaccination regimens. Thus, double immunization with all tested vaccines more efficiently activated proliferation of lymphocytes in lymphoid organs of experimental animals compared with single vaccination (Figure 6, Appendix A). Moreover, the number of vaccinations affects the expression of immune checkpoints in splenocytes. In most cases, double vaccination with DC-based and liposomal vaccines resulted in down-regulation of both positive and negative immune checkpoints compared with a single dose, whereas booster immunization with CIMV-based vaccines, in contrast, up-regulated the expression of CD27 and 4-1BB-L (positive) as well as CTLA-4 and TIM3 (negative) immune checkpoints (Figure 7). We can assume that the missing effect of the booster vaccination can be associated with the untimely restimulation of the immune response. Probably, a 7-day interval between the vaccines is not optimal and should be reduced to 3–4 days. This problem needs further investigation.

The main positive immunologic result of the vaccination of tumor-bearing mice with vaccines under the study was the activation of Th1/Th17 immune response. Generally, antitumor DC-based vaccines frequently activate the Th1/Th17 response, which correlates well with the strong antitumor activity of the vaccines [69,70,71,72,73]. It is known that Th17 cells modulate inflammation, are negatively correlated with Tregs [74,75], and play a pivotal role in the activation of tumor-specific CD8+ T cells [76]. Adoptive transport of tumor-specific Th17 cells was demonstrated to retard tumor growth and improve survival in murine tumor models [75,77]. Thus, protective inflammation elicited by Th17 cells promotes the activation of antitumor immunity and helps Th1 to stimulate tumor-specific CD8+ T cells. Interestingly, CIMV-based and liposomal vaccines also activated Th1/Th17 cells, indicating apparently similar mechanisms of action of the vaccines to DCs. We suppose that the source of tumor antigens to load DCs determines to a greater extent the activation of the Th17 cells after the vaccination. Indeed, a whole tumor cell set of antigens was used both in our study (total tumor RNA) and in other studies (tumor cell lysates [69,73], irradiated tumor cells [71,72], DC/tumor cell hybrids [70]) where Th17 response was polarized after the vaccination. It is possible that self-antigens contained in such sources of antigens can trigger autoimmune reactions that play an important role in the activation of Th17 cells.

Thus, we have demonstrated that non-cellular vaccines based on CIMVs or liposomes have a high antitumor potential and can efficiently activate antitumor immunity that results in significant retardation of tumor growth and inhibition of metastasis development. Among the vaccines under the study, CIMV-based vaccines seem to be the most promising and provide the basis for developing a novel highly efficient antitumor immunotherapeutic approach.

## 5. Conclusions

In this study, we demonstrated that cell-free CIMV-based vaccines exhibited superior antitumor and antimetastatic activity in a tumor model in vivo. CIMV-based vaccines stimulated CTLs in vivo that highly efficiently lysed tumor cells. The antitumor activity of the different types of the vaccines was similar: they exhibited similar ability to decrease the sizes of primary tumor nodes and the number of metastases regardless of the vaccine type in the case of single vaccination, and the booster vaccination did not improve the overall antitumor efficacy. CIMV-based vaccine application resulted in switching of humoral immune response formed upon RLS_40_ progression in mice to cellular immunity by activation of Th1/Th17 cells as well as induction of positive immune checkpoint 4-1BBL and downregulation of suppressive immune checkpoints in a raw PD-1 >>> TIGIT > CTLA4 > TIM3.

## Figures and Tables

**Figure 1 pharmaceutics-14-02542-f001:**
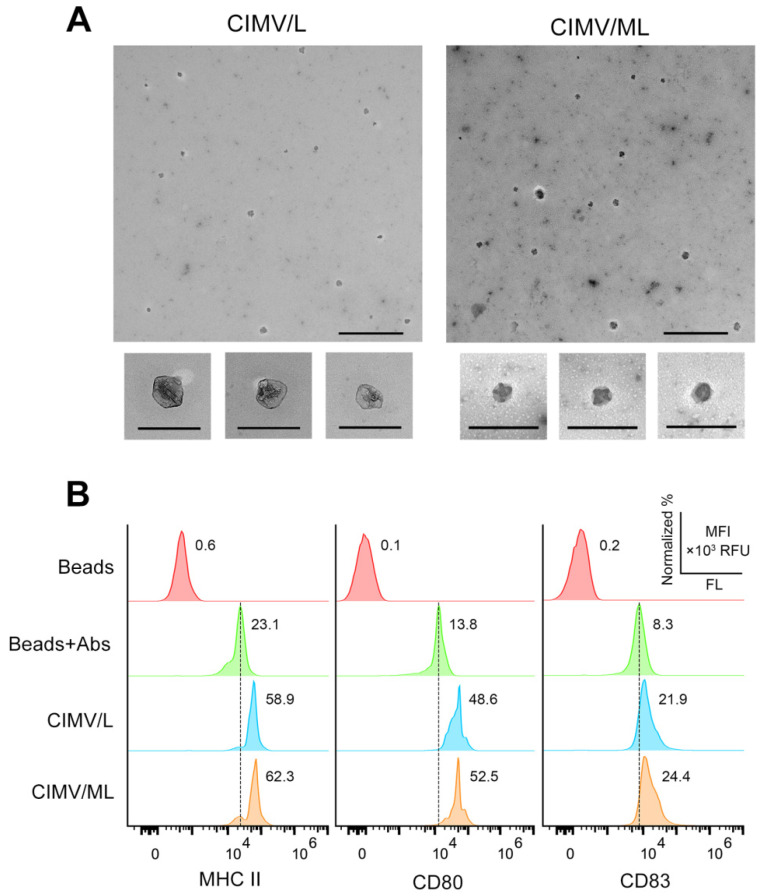
Characterization of tsDC-derived CIMVs. (**A**) Analysis of the morphology of CIMVs using TEM. TEM data are presented as microphotographs with scale bars = 500 nm (upper pictures) and 200 nm (bottom small pictures). (**B**) Surface expression of MHC II, CD80, and CD83 on DC-derived CIMVs. CIMVs were immobilized on aldehyde/sulfate latex beads, blocked with glycine, stained with anti-MHC II (FITC), anti-CD80 (APC), and anti-CD83 (PE) antibodies, and analyzed by flow cytometry. Data are presented as distribution of CIMVs by mean fluorescence intensity (MFI).

**Figure 2 pharmaceutics-14-02542-f002:**
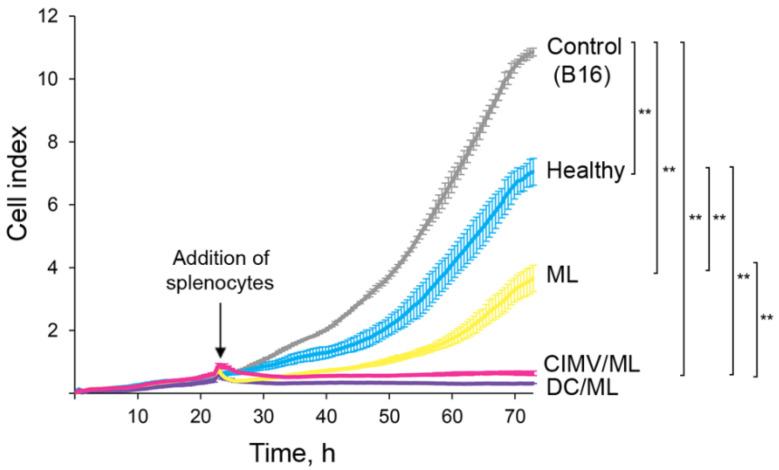
B16-specific cytotoxicity of splenocytes induced in vivo by the vaccines under the study. Control—B16 cells incubated in the absence of splenocytes (gray line). Healthy—B16 cells incubated with splenocytes from healthy mice (blue line). ML, CIMV/ML, and DC/ML corresponded to B16 cells incubated with splenocytes isolated from mice vaccinated with ML, CIMV/ML, or DC/ML vaccines, respectively (yellow, magenta, and indigo lines, respectively). The viability of B16 cells was assessed in real-time mode using xCELLigence (ACEA Biosciences, San Diego, CA, USA). Data presented as mean ± SD. Data were statistically analyzed using one-way ANOVA with post hoc Tukey test; *p*-value indicates a statistically reliable difference: ** *p* < 0.01.

**Figure 3 pharmaceutics-14-02542-f003:**
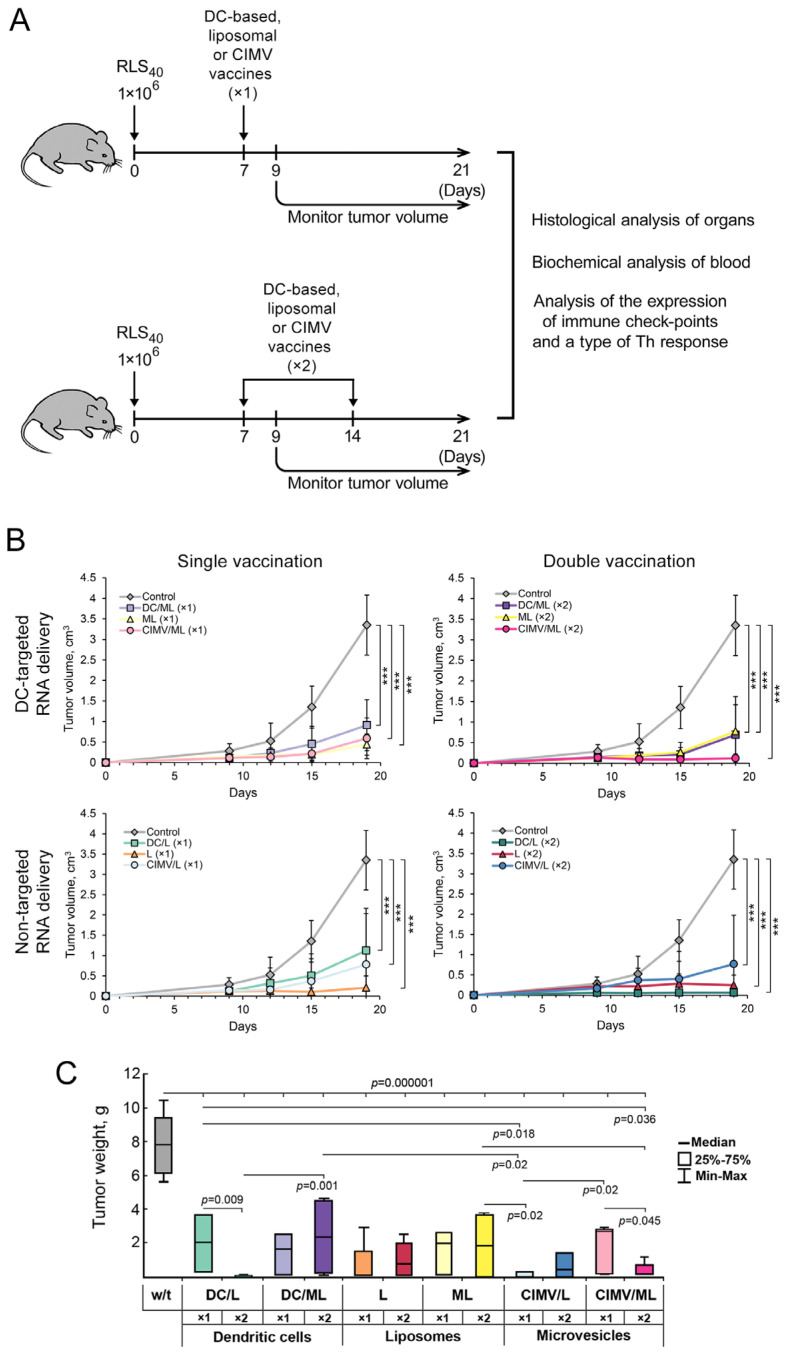
Effect of DC-based and non-cellular vaccines on the growth of RLS_40_ tumor in vivo. (**A**) Experimental design. RLS_40_ cells (1 × 10^6^) were implanted i/m into CBA/LacSto mice. Animals received vaccines s/c on day 7 (single vaccination) or days 7 and 14 (double vaccination). (**B**) Tumor growth curves in groups of RLS_40_-bearing mice treated once or twice with DC-, liposome-, or vesicle-based vaccines. Vaccines were prepared using DC-targeted mannosylated liposomes ML or non-targeted cationic liposomes L. Data are presented as mean ± S.D. Data were statistically analyzed using one-way ANOVA with post-hoc Fisher test; *p*-value indicates a statistically reliable difference: *** *p* < 0.0005. (**C**) Tumor weight of experimental animals on day 21 of tumor growth. Data are represented as median. Data were statistically analyzed using one-way ANOVA with post-hoc Fisher test, *p*-values indicate a statistically reliable difference. Color codes correspond to Table 2.

**Figure 4 pharmaceutics-14-02542-f004:**
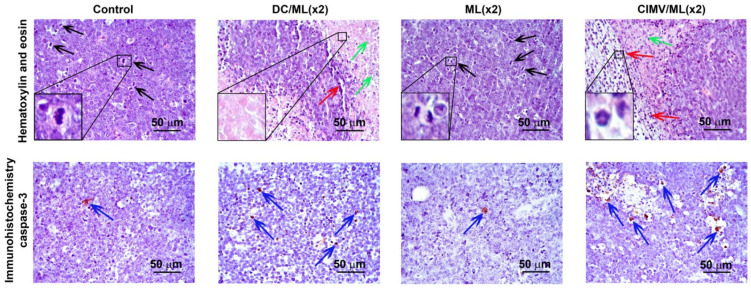
Vaccines inhibit mitotic activity and induce apoptosis in RLS_40_ lymphosarcoma tissue. Representative histological images of the tumor sections. Hematoxylin and eosin staining (upper panel) and immunohistochemical staining with anti-caspase-3 primary antibodies (bottom panel). Black arrows indicate mitosis events. Red arrows indicate inflammatory infiltration. Green arrows indicate necrotic changes. Blue arrows indicate caspase-3-positive cells. Black boxes indicate typical examples of mitosis, necrosis, and granulocyte. Original magnification ×400.

**Figure 5 pharmaceutics-14-02542-f005:**
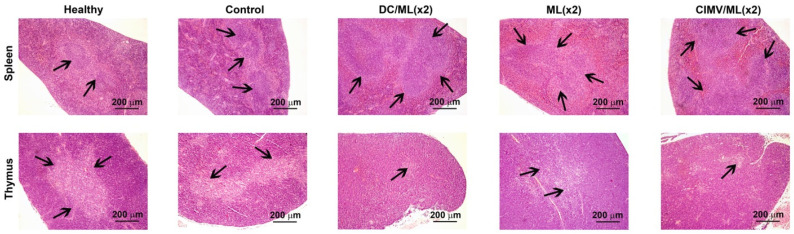
Immunomodulatory effects of vaccines in RLS_40_-bearing mice. Representative histological images of the spleen (**upper panel**) and thymus (**bottom panel**) sections. Hematoxylin and eosin staining. Black arrows indicate lymphoid follicles in the spleen (**upper panel**) and medulla in the thymus (**bottom panel**). Original magnification ×100.

**Figure 6 pharmaceutics-14-02542-f006:**
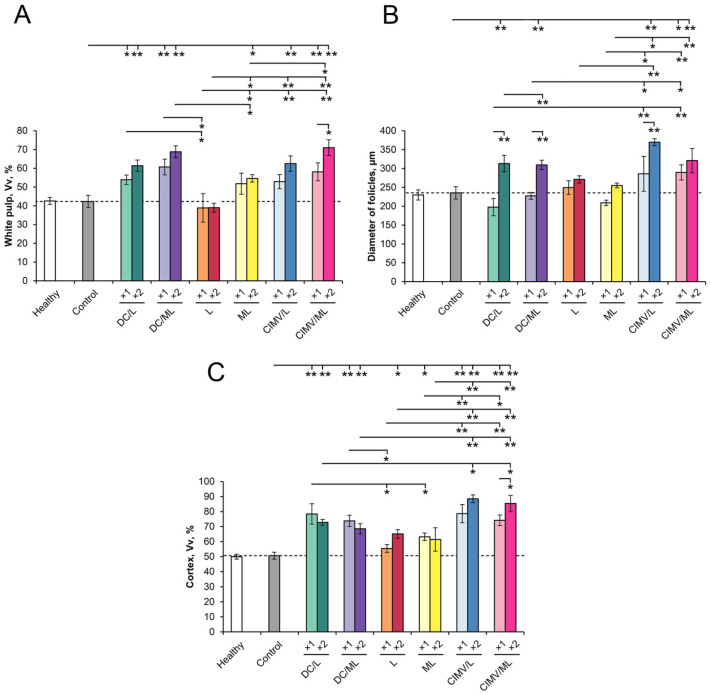
Spleen and thymus morphometry of RLS_40_ lymphosarcoma-bearing mice without treatment (control) and after vaccine administration. (**A**) Volume density of white pulp in the spleen. (**B**) Diameter of lymphoid follicles in the spleen. (**C**) Volume density of the cortex in the thymus. Dash lines represent the levels of the control group. Data are presented as mean ± S.E.M. Data were statistically analyzed using one-way ANOVA with post-hoc Fisher test. Statistically reliable differences between the groups: * *p* < 0.05, ** *p* < 0.005.

**Figure 7 pharmaceutics-14-02542-f007:**
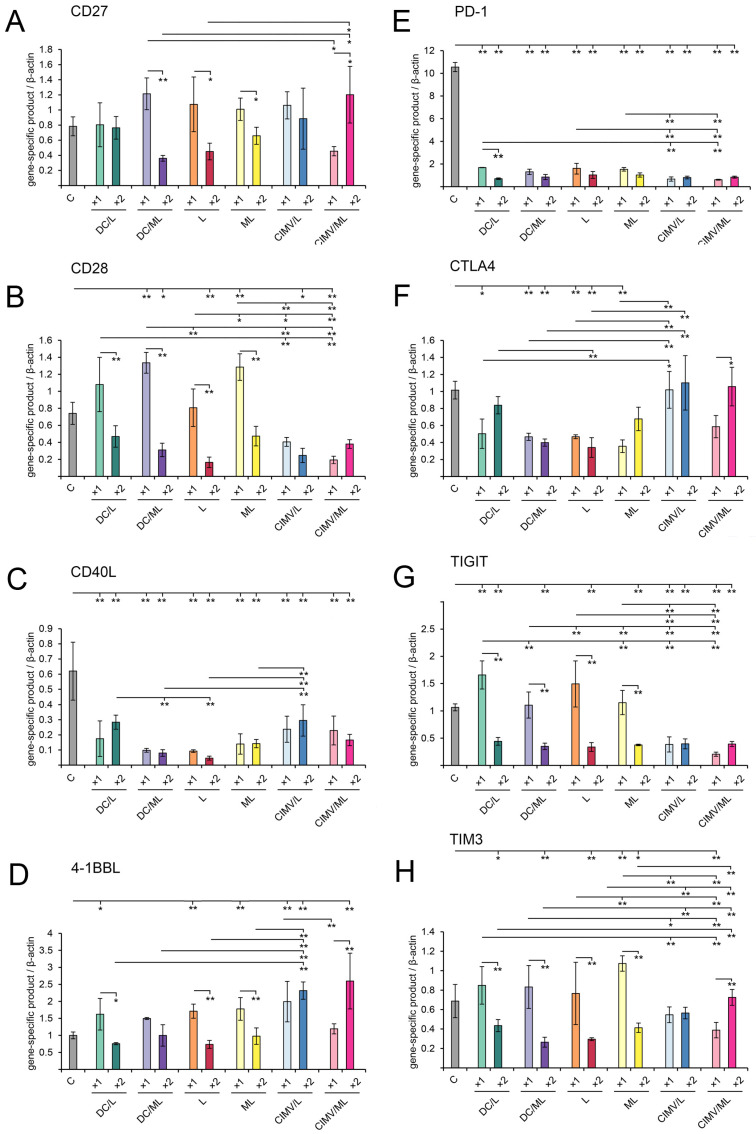
The expression levels of positive (**A**–**D**) and negative (**E**–**H**) immune checkpoints in splenocytes of mice with RLS_40_ after immunization with DC-based or cell-free vaccines determined by qPCR and TaqMan probes. The expression level of gene-specific product was normalized to the expression level of β-actin. The expression levels of genes are presented for the 21st day after tumor implantation. Data are presented as mean ± S.E.M. Data were statistically analyzed using one-way ANOVA with post-hoc Tukey test. Statistically reliable differences between the groups: * *p* < 0.05, ** *p* < 0.01.

**Figure 8 pharmaceutics-14-02542-f008:**
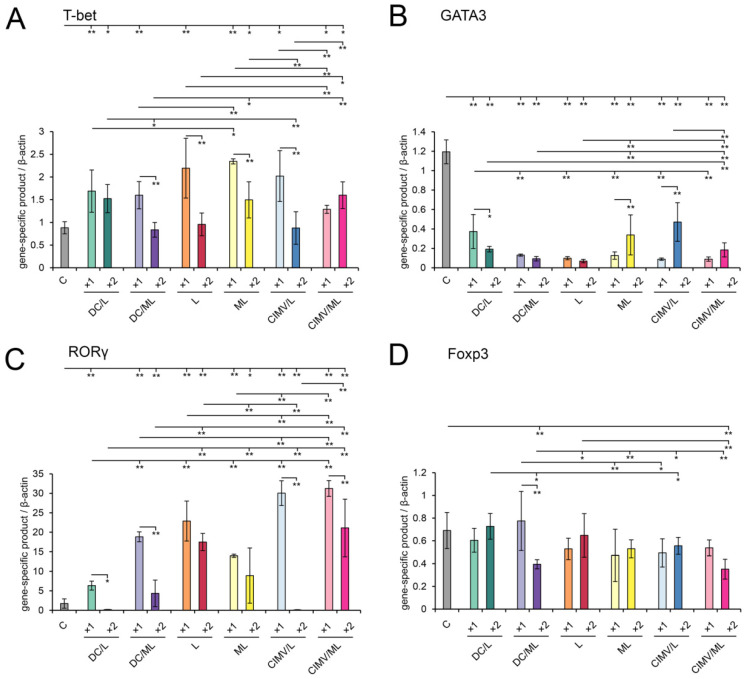
The expression levels of master regulators of T helper immune response (**A**) T-bet, (**B**) GATA3, (**C**) RORγ, and (**D**) Foxp3 in splenocytes of RLS_40_-bearing mice treated with DC-, liposome-, and CIMV-based vaccines determined by qPCR and TaqMan probes. The expression level of gene-specific product was normalized to the expression level of β-actin. The expression levels of genes were presented on the 21st day after tumor transplantation. Data are presented as mean ± S.E.M. Data were statistically analyzed using one-way ANOVA with post-hoc Tukey test. Statistically reliable differences between the groups: * *p* < 0.05, ** *p* < 0.01.

**Table 1 pharmaceutics-14-02542-t001:** List of primers and probes used in the study.

Gene	Sequences of Primers and Probes, 5′ → 3′	Amplicon Size, bp ^1^
CD27	F ^2^: TTGTGACCTTCTCCAGCATG R: GTAAGGACAAGGCTCTTCAGG Probe: ((5,6)-FAM)-AAGAACAAGATTGCACCCAGGACGA-BHQ1	122
CD28	F: GTTGCTGGAGTCCTGTTTTG R: CTGGTAAGGCTTTCGAGTGAG Probe: ((5,6)-FAM)-ATGGCTTGCTAGTGACAGTGGCT–BHQ1	147
CD40L	F: GTCAGCATGATAGAAACATACAGC R: TGGGTGATAAGGAAAACAGTAAGT Probe: ((5,6)-FAM)-CCCCCAGATCCGTGGCAACT–BHQ1	110
4-1BBL	F: TGCCCCAACACTACACAAC R: TCTTCGTACCTCAGACCTTGA Probe: ((5,6)-FAM)-CTCTCCTGTGTTCGCCAAGCTACT–BHQ1	146
PD1	F: GGTACCCTGGTCATTCACTTG R: ATTTGCTCCCTCTGACACTG Probe: ((5,6)-FAM)-ACCTCTAGAAGCCACCCTGATTGC–BHQ1	133
CTLA4	F: TCTGCAAGGTGGAACTCATG R: AGCTAACTGCGACAAGGATC Probe: ((5,6)-FAM)-ATAAATCTGCGTCCCGTTGCCCA–BHQ1	132
TIGIT	F: CTGTAGGCCTCTGGTTAGAAG R: TGACAGAGCCACCTTCCT Probe: ((5,6)-FAM)-CAGCCTGTATCAGCCCCTGGAC–BHQ1	149
TIM3	F: ACCCTGGCACTTATCATTGG R: TTTTCCTCAGAGCGAATCCTG Probe: ((5,6)-FAM)-TCCTGCATTTGCCAACCCTCCT–BHQ1	149
T-bet	F: TTCAACCAGCACCAGACAG R: AGACCACATCCACAAACATCC Probe: ((5,6)-FAM)-TCACTAAGCAAGGACGGCGAATGT–BHQ1	124
GATA3	F: ACCTCACCACCCTTCCA R: TTCATGATACTGCTCCTGCG Probe: ((5,6)-FAM)-CTCCGACCCCTTCTACTTGCGTTTT–BHQ1	141
RORγ	F: TTTCTGAGGATGAGATTGCCC R: TTGTCGATGAGTCTTGCAGAG Probe: ((5,6)-FAM)-CCAGGACGGTTGGCATTGATGAGA–BHQ1	146
Foxp3	F: AAGTACCACAATATGCGACCC R: TCTGAAGTAGGCGAACATGC Probe: ((5,6)-FAM)-TCACCTATGCCACCCTTATCCGATG–BHQ1	132
β-actin	F: TATTGGCAACGAGCGGTTCC R: TGGCATAGAGGTCTTTACGG Probe: ((5,6)-ROX)-CCAGCCTTCCTTCTTGGGTATGGAATCC–BHQ2	140

^1^ bp—base pairs; ^2^ F—forward primer, R—reverse primer.

**Table 2 pharmaceutics-14-02542-t002:** Vaccine types and composition, list of experimental groups and regimens of vaccination of RLS_40_-bearing mice.

Type	Name ^1^	Color Code ^2^	Vaccine Composition	Vaccination Regimen ^3^	Dose, per Mouse
Cellular vaccines	DC/L(×1)	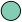	DCs transfected with RNA-RLS_40_/L	d7	10^5^ cells
DC/L(×2)	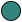	d7, d14
DC/ML(×1)	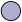	DCs transfected with RNA-RLS_40_/ML	d7
DC/ML(×2)	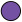	d7, d14
Liposomal vaccines	L(×1)	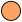	RNA-RLS_40_/L lipoplexes	d7	5 µg RNA
L(×2)	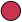	d7, d14
ML(×1)	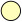	RNA-RLS_40_/ML lipoplexes	d7
ML(×2)	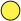	d7, d14
Vesicular vaccines	MV/L(×1)	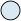	CIMVs isolated from DCs transfected with RNA-RLS_40_/L	d7	15 µg (total protein count)
MV/L(×2)	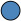	d7, d14
MV/ML(×1)	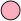	CIMVs isolated from DCs transfected with RNA-RLS_40_/ML	d7
MV/ML(×2)	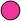	d7, d14

^1^ The same designation was used for the experimental groups. ^2^ Color code was used throughout the article to show data corresponded to a particular group. ^3^ Day(s) from RLS_40_ lymphosarcoma transplantation.

**Table 3 pharmaceutics-14-02542-t003:** Morphometry of RLS_40_ lymphosarcoma tissue of mice without treatment (control) and after vaccine administration.

Groups	Color Code	Mitotic Cells, Nv	Caspase-3 Positive Cells, Nv
Control	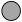	5.6 ± 0.5	1.4 ± 0.1
DC/L(×1)	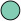	4 ± 0.7 *	1.6 ± 0.1
DC/L(×2)	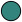	**2 ± 0.2 ***	**6.6 ± 0.9 ***
DC/ML(×1)	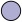	2.6 ± 0.8 *	1.9 ± 0.5
DC/ML(×2)	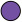	**1.7 ± 0.4 ***	**4.8 ± 0.4 ***
L(×1)	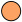	4 ± 1.5	2.2 ± 0.4
L(×2)	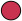	5.4 ± 1.5	1.3 ± 0.1
ML(×1)	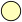	4.1 ± 0.5 *	0.9 ± 0.4
ML(×2)	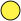	5.2 ± 1	1.4 ± 0.1
CIMV/L(×1)	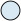	4.2 ± 0.2 *	1.5 ± 0.5
CIMV/L(×2)	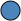	**2.5 ± 0.6 ***	**6.3 ± 0.4 ***
CIMV/ML(×1)	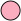	5 ± 0.7	2.6 ± 0.8 *
CIMV/ML(×2)	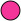	**2.1 ± 0.7** *****	**5.8 ± 1.4 ***

* Differences from the control mice were statistically significant at *p* ≤ 0.05. The largest alterations are indicated in bold.

## Data Availability

Not applicable.

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
