# Peer review of "Dendritic Cell-Derived Artificial Microvesicles Inhibit RLS40 Lymphosarcoma Growth in Mice via Stimulation of Th1/Th17 Immune Response"

_pharmaceutics, 2022, doi:10.3390/pharmaceutics14112542_

Round 1

Reviewer 1 Report

The authors Markov et.al has shown here the usage of cell free antitumor vaccines, In there study they have mainly used three types of vaccine -Cellular DC based vaccine, Liposome based vaccine and Vesicular vaccine (CIMVs). The authors have done a beautiful and article is overall well written. here they have shown that Liposome or CMV derived vaccine efficiently activated antitumor CTLs in in vivo. Later they have shown the mice vaccinated with Liposome and CIMV based vaccine showed significant retardation of growth in primary tumor node.  Later in histological analysis they have shown further that CIMV and DC based vaccines have decreased mitotic activity and has more immunomodulatory activity in spleen.  Further they have analyzed the genes involved in which are regulators of immune response and Th cell development and found that vaccination resulted in activation of Th1/TH7 immune check points.

Author Response

Thank you very much for your valuable review of our work.

Reviewer 2 Report

In the research article “Dendritic Cell-Derived Artificial Microvesicles Inhibit RLS40 2 Lymphosarcoma Growth in Mice via Stimulation of Th1/Th17 3 Immune Response” Markov, O.V. et al., have compared the antitumor potential of cell-free vaccines developed based on microvesicles derived from dendritic cells (DCs) with DC- and cationic liposome-based vaccines using murine model of drug-resistant lymphosarcoma RLS40 in vivo.  Authors have reported that  the developed vaccines (CIMV and DC vaccines) are not hepatotoxic and when administered subcutaneously, exhibited comparable potential to stimulate highly efficient antitumor CTLs in vivo.  Unlike CIMV and DC vaccines, the liposomal vaccines are 25% weaker CTL inducers.  Despite these differences in the induction of CTLs, the antitumor efficiencies of all these vaccines reported to be similar.  For example the effect on reducing the size of tumor nodes and the number of liver metastases, which were significantly decreased, regardless of the vaccine type.   The manuscript is well written and the authors have interpreted the obtained results as appropriate.  The discussion, although lengthy, and the conclusions are as appropriate

Minor comments

1.     In materials and methods section

Table-1: Add the product length and mention the reference from which the primer sequences have been taken

Section 2.5.1:  What is the efficiency of transfection with L and ML formulations

2.     The discussion is too long.  Authors are requested to reduce the discussion and restrict to a max of 2 pages.  Currently it is about 3 pages

3.     Authors would have tested the stability of prepared samples by comparing the safety and efficacy profiles

Author Response

Thank you very much for your valuable review of our work.

Minor comments

1) In materials and methods section

1.1) Table-1: Add the product length and mention the reference from which the primer sequences have been taken

Product lengths were added to Table 1. Tools used for primer/probe design were added to Section 2.12:

To design primers and probes, we utilized RealTime PCR Tool (https://eu.idtdna.com/scitools/Applications/RealTimePCR/). The secondary structure of the primers/probes and the self- and heterodimerization tendencies of each primer/probe set were predicted using OligoAnalyzer Tool (https://eu.idtdna.com/calc/analyzer). Primers and probes were synthesized in the Laboratory of Biomedical Chemistry of ICBFM SB RAS (Novosibirsk, Russia).

1.2) Section 2.5.1:  What is the efficiency of transfection with L and ML formulations

Added to Section 2.5.1:

Transfection efficiency of L and ML liposomes was 40-50% of DCs (Markov O. et al. Journal of Controlled Release, 2015).

2) The discussion is too long.  Authors are requested to reduce the discussion and restrict to a max of 2 pages.  Currently it is about 3 pages

Corrected. We reduced the discussion as much as possible. We kept important parts so that the reader can interpret the results of our investigation more deeply.

  1. Authors would have tested the stability of prepared samples by comparing the safety and efficacy profiles.

We fully agree with you that vaccine stability studies are very important. However, a large amount of performed work did not allow us to include vaccine stability studies in the present manuscript. In our future investigations of CIMV-based vaccines, we will definitely investigate stability of both cellular and non-cellular vaccines in vitro and in vivo.

Reviewer 3 Report

The authors compare three different anti-tumor vaccines (DCs loaded with tumor RNA, Liposomes carrying tumor RNA and CIMVS derived from DCs loaded with tumor RNA using the murine model of drug-resistant lymphosarcoma. They provide promising approaches to optimize immunotherapy with the aim to replace the complicated manufacturing process of cellular based vaccines by cell-free vaccines. The use of CIMVs as verhicles delivering compounds or parental cell content has only recently been described and characterized. Its application in this study is a novel and very interesting especially its comparison to the liposome and DC based more standard vaccines.  Ex vivo and in vivo induction of differential antitumor immune responses (TH1/TH1) have been demonstrated and important clinical relevant aspects as impact of administration routes, (liver) toxicities have been analyzed and discussed. An explanation/hypothesis for the missing effect of the booster vaccination would be beneficial. Is there any information available about immunological differences of the CIMVS derived from RNA vs. tumor lysate loaded DCs?

The work is highly relevant and the paper is well written and therefore highly recommended to be published in your journal.

Some spelling mistakes should be corrected (e.g. abstract last sentence)

Author Response

Thank you very much for your valuable review of our work.

1) An explanation/hypothesis for the missing effect of the booster vaccination would be beneficial.

Corrected. We added our thoughts on this topic to Discussion section:

We can assume that missing effect of the booster vaccination can be associated with untimely restimulation of the immune response. Probably, a 7-day interval between the vaccines is not optimal and should be reduced to 3-4 days. Further investigations should be performed.

2) Is there any information available about immunological differences of the CIMVS derived from RNA vs. tumor lysate loaded DCs?

It is very interesting question. Antitumor and immunostimulatory potential of DC-derived CIMVs is quite a novel research area. To date, unfortunately, there is no any information about immunological differences between the CIMVs derived from tumor lysate-loaded DCs vs RNA transfected DCs. However, in the most recent work (Kim et al. Vaccines 2022, 10(11), 1877; https://doi.org/10.3390/vaccines10111877) it was demonstrated that artificial APC (HEK293T cell stably expressed HLA and co-stimulatory molecules) loaded with pp65 model antigen peptide and CIMVs derived from them had similar T-cell-stimulating effect, whereas APCs endogenously expressing pp65 peptide (after lentiviral transduction) possessed significantly higher T-cell stimulatory potential compared with CIMVs derived from them. In other words, Kim et al. have demonstrated that the source of antigen is essential for immunostimulatory properties of APC-derived CIMVs – exogenous antigen is preferable over endogenously translated and processed one. Whether DCs loaded with tumor lysate (exogenous tumor antigens) would be a more immunogenic source of CIMVs compared with DCs transfected with tumor RNA (endogenous tumor antigens expression) is a big question. Additional investigations should be performed.

3) Some spelling mistakes should be corrected (e.g. abstract last sentence)

Corrected.